# Therapeutic Benefits of Stem Cells and Exosomes for Sulfur-Mustard-Induced Tissue Damage

**DOI:** 10.3390/ijms24129947

**Published:** 2023-06-09

**Authors:** Carol Christine Bosholm, Hainan Zhu, Pengfei Yu, Kun Cheng, Sean Vincent Murphy, Patrick Michael McNutt, Yuanyuan Zhang

**Affiliations:** 1Wake Forest Institute for Regenerative Medicine, Wake Forest University Health Sciences, Winston-Salem, NC 27101, USA; cbosholm@wakehealth.edu (C.C.B.); hzhu@wakehealth.edu (H.Z.); pyu@wakehealth.edu (P.Y.); semurphy@wakehealth.edu (S.V.M.); pmcnutt@wakehealth.edu (P.M.M.); 2Division of Pharmacology and Pharmaceutical Sciences, School of Pharmacy, University of Missouri-Kansas City, 2464 Charlotte Street, Kansas City, MO 64108, USA; chengkun@umkc.edu

**Keywords:** exosomes, stem cells, therapeutic impact, sulfur mustard, skin, wound healing

## Abstract

Sulfur mustard (SM) is a highly toxic chemical agent that causes severe tissue damage, particularly to the eyes, lungs, and skin. Despite advances in treatment, there is a need for more effective therapies for SM-induced tissue injury. Stem cell and exosome therapies are emerging as promising approaches for tissue repair and regeneration. Stem cells can differentiate into multiple cell types and promote tissue regeneration, while exosomes are small vesicles that can deliver therapeutic cargo to target cells. Several preclinical studies demonstrated the potential of stem cell, exosome, or combination therapy for various tissue injury, showing improvements in tissue repairing, inflammation, and fibrosis. However, there are also challenges associated with these therapies, such as the requirement for standardized methods for exosome isolation and characterization, the long-term safety and efficacy and reduced SM-induced tissue injury of these therapies. Stem cell or exosome therapy was used for SM-induced eye and lung injury. Despite the limited data on the use for SM-induced skin injury, this therapy is a promising area of research and may offer new treatment options in the future. In this review, we focused on optimizing these therapies, evaluating their safety and efficacy, and comparing their efficacy to other emerging therapeutic approaches potentially for SM-induced tissue injury in the eye, lung, and skin.

## 1. Backgrounds

The development of innovative medical therapies to safeguard civilians against both intentional and accidental toxic chemical exposure is of paramount importance. Vesicants, including anticancer drugs (i.e., nitrogen mustard for lymphoma) and chemical warfare agents (i.e., sulfur mustard [SM], Lewisite, and phosgene oxime), often induce local and systemic toxicity [1]. SM is a potent chemical warfare agent that was used in several conflicts throughout history. SM can cause acute, subacute, chronic, or severe injuries in the eye, lung, and skin, predisposing casualties to secondary deleterious effects, scar formation, and reduced quality of life. Despite advances in treatment, there are no effective therapies for SM-induced tissue injury due to inflammation and basal cell damage [2]. There is an urgent need for researchers to develop novel and effective therapies to improve the outcomes of patients suffering from SM-induced injury. The use of stem cell and exosome therapies showed promising results in preclinical studies and has the potential as a novel therapeutic strategy for this condition [3,4,5,6,7,8,9].

Stem cells are undifferentiated cells that can differentiate into multiple cell types and promote skin tissue regeneration. Several types of mesenchymal stem cells (MSCs), including bone-marrow-derived stem cells (BMSCs), amniotic-fluid-derived stem cells (AFSCs), umbilical-cord-derived stem cells (UCSCs), placental-derived stem cells (PSCs), adipose-derived stem cells (ASCs), and urine-derived stem cells (USCs) (Table 1), were investigated for their potential in treating skin injury [10,11,12,13,14,15,16,17,18,19,20]. Preclinical studies showed that stem cell therapy can improve wound healing, reduce inflammation, and decrease fibrosis in animal models of skin injury. Stem cell therapy was effective in SM-induced lung injury by alleviating inflammation and promoting tissue repair [21]. Exosomes are small vesicles that are secreted by cells and can deliver therapeutic cargo to target cells. Exosomes were investigated for treating tissue injury due to their ability to modulate inflammation, promote tissue repair, and enhance wound healing [22]. Exosome therapy improves tissue repairing, reduces inflammation, and decreases fibrosis in animal models of skin injury [7,23]. However, there are still several challenges associated with stem cell or exosome therapy, such as the need for standardized methods for cell or exosome isolation and characterization, as well as concerns about the long-term safety and efficacy of the approach [24,25].

Combining MSCs and exosome therapy may provide a synergistic effect for treating SM-induced tissue injury. Stem cells can provide a source of regenerative cells and promote tissue repair, while exosomes can deliver therapeutic cargo to target cells and modulate the immune response. Preclinical studies showed that combining stem cell and exosome therapy enhances wound healing by reducing inflammation and fibrosis in animal models of surgically created full thickness skin wound [20,26].

## 2. Incidence Rate of SM-Induced Tissue Injury

SM is a potent alkylating agent that causes severe tissue injury in the eyes, lungs, and skin through direct cytotoxicity and oxidative stress [2,27,28,29].

The incidence rate of SM-induced tissue injury varies depending on the exposure scenario. Military conflicts were the primary cause of SM exposure, with the highest incidence rates reported during the Iran–Iraq war in the 1980s. According to a study published in 2019, the prevalence of SM-induced tissue injuries among Iranian veterans was 42.3%, with most cases being mild or moderate in severity [30,31].

Occupational exposure to SM can also occur in industries involved in the production of chemical weapons, and cases of SM-induced tissue injury were reported among workers in these industries [29]. Although SM was banned under the Chemical Weapons Convention, there is still a risk of exposure to this agent in certain regions due to the presence of old stockpiles and abandoned weapons [32].

## 3. Pathophysiology and Complications of Sulfur-Mustard-Induced Tissue Injury

The pathophysiology of SM-induced tissue injury involves several mechanisms, including direct cytotoxicity, oxidative stress, inflammation, and immunosuppression. SM-induced skin damage is used as an example (see Figure 1).

The primary mode of SM toxicity is formation of inter-strand DNA crosslinks. At higher doses, lower-affinity interactions such as protein/lipid binding and ROS contribute to cell death [32]. SM causes direct damage to skin cells by forming covalent bonds with proteins and other biomolecules, leading to cell death. This process disrupts the epidermis and dermis layers, causing blistering and necrosis. In addition, SM induces the production of reactive oxygen species (ROS) that can damage cellular membranes, proteins, and DNA. The accumulation of ROS also activates pro-inflammatory pathways and promotes apoptosis of skin cells. Moreover, SM-induced injury triggers the release of pro-inflammatory cytokines [33,34,35], such as interleukin-1 (IL-1) and tumor necrosis factor-alpha (TNF-α), leading to an influx of inflammatory cells. The inflammatory response contributes to the formation of blisters and tissue damage. Furthermore, SM exposure can also cause immunosuppression by inhibiting the function of immune cells [36], such as T and B cells [37], and suppressing the production of cytokines involved in the immune response. Importantly, SM binds constantly to skin tissues within several minutes after contact if decontamination is not performed immediately after exposure [38]. This injury induces basal cells damage and further exhausts the local stem cell pool [39,40], leading to irreversible cutaneous injury. 

SM-induced skin injury is a complex process involving multiple mechanisms. The severity and extent of skin injury depend on the concentration and duration of exposure to SM, as well as the individual’s susceptibility and the treatment received.

The acute symptoms of SM-induced skin injury include pain, itching, erythema, blistering, and edema. These symptoms can be severe and require prompt medical attention to prevent secondary infections [28]. In addition, SM exposure can also lead to long-term complications, including scarring, hyper-pigmentation, and delayed wound healing. Chronic skin changes may take several months or years to develop and can be associated with psychological distress and decreased quality of life [40].

Furthermore, SM exposure was linked to an increased risk of developing various cancers, including skin cancer, lung cancer, and hematological malignancies [31]. The carcinogenic potential of SM is believed to result from the induction of oxidative stress, DNA damage, mutagenesis due to the DNA alkylation that are not correctly repaired, and inhibition of DNA repair mechanisms.

## 4. Current Management and Treatment Options for SM-Induced Tissue Injury

The principles of treatment for SM-induced tissue injury include immediate decontamination to prevent further exposure, wound care to promote healing and prevent infection, pain management, and rehabilitation to minimize functional impairment [28].

To prevent and treat an SM-induced skin injury, for example, key steps in treatment include: (i), decontamination is first needed. The affected area should be washed with soap and water as soon as possible after exposure to remove any remaining SM from the skin; (ii) wound care should be initiated as soon as possible after exposure to prevent infection and promote healing. The wound should be cleaned and dressed regularly, and topical antibiotics or corticosteroids may be applied to reduce inflammation and prevent infection; (iii) pain management is an important part of treatment and can be achieved with NSAIDs or opioid analgesics; (iv) physical therapy may be necessary to prevent contractures and improve range of motion. This may include exercises, splinting, and massage; and (v) psychological support is required for the patients. SM-induced skin injury can cause significant psychological distress, and patients may require counseling or other forms of psychological support to cope with the long-term effects of the injury.

Research is ongoing to develop more effective treatments for SM-induced injury, including the use of growth factors, stem cell therapy, and other novel therapies [41].

Currently, there is no specific treatment for SM-induced skin injury, and management is primarily supportive. The goal of treatment is to minimize pain, prevent infection, and promote healing of the affected area [42]. Some of the commonly used treatments are as follows: (i) debridement: removal of dead or necrotic tissue from the affected area; (ii) topical agents: application of topical corticosteroids, antibiotics, or silver sulfadiazine to prevent infection and promote healing; (iii) dressings: application of occlusive or semi-occlusive dressings to reduce inflammation and promote wound healing; (iv) pain management: use of nonsteroidal anti-inflammatory drugs (NSAIDs) or opioid analgesics to manage pain; (v) surgical intervention: in severe cases, surgical intervention may be necessary to remove necrotic tissue and promote healing; (vi) rehabilitation: use of physical therapy to prevent contractures, improve range of motion, and promote wound healing.

Several experimental treatments were investigated for SM-induced tissue injury, including hyperbaric oxygen therapy, growth factors, and stem cell transplantation, but none were proven effective in clinical trials [43].

## 5. Challenges of Current Treatment for SM-Induced Tissue Injury

Although several treatment options are available for SM-induced tissue injury, there are several challenges associated with these treatments. Some of the major challenges include delayed diagnosis, inadequate wound care, and lack of effective treatments for chronic SM-induced tissue injury.

Using SM-induced-skin injury as an example, delayed diagnosis is a major challenge in the treatment because symptoms may not appear for several hours or even days after exposure. In addition, the diagnosis of SM-induced skin injury can be challenging, as it can be difficult to distinguish from other skin conditions. This delay in diagnosis can lead to inadequate treatment, which can worsen the severity of the injury [31].

Inadequate wound care is another challenge in the treatment of SM-induced skin injury. Effective wound care is critical to prevent infection and promote healing. However, inadequate wound care can lead to chronic wounds, scarring, and functional impairment. In addition, some of the treatments used for skin injury, such as silver sulfadiazine, may have side effects that delay or impair wound healing [40].

Finally, currently, neither known antidote nor effective treatment exists for sulfur mustard exposure. Despite the use of several experimental treatments, including hyperbaric oxygen therapy, growth factors, and stem cell transplantation, none of these treatments were proven effective in clinical trials [29].

## 6. Stem Cell and Exosome Therapy for Tissue Injury

Stem cells and exosomes display promising therapeutic effects for skin injury, with ongoing research and clinical trials exploring their potential [44]. Stem cells, due to their unique ability in cell replacement and cytokines/immune modulation, promote skin regeneration and wound healing. Exosomes, small extracellular vesicles secreted by stem cells, also gained attention for their potential to enhance tissue repair and regeneration through paracrine signaling [45,46]. The unique properties of stem cells and exosomes or extracellular vesicles are immunomodulatory, anti-inflammatory, antifibrotic, antioxidative, and antiapoptotic effects for the treatment of tissue injury [23,47]. While these therapies are still in the early stages of development and more research is needed, they hold great promise for improving skin injury treatment outcomes in the treatment of various skin wounds, including SM-induced tissue injury. 

Using stem cell therapy for skin injury as an example, stem cells can be sourced from the patient’s own body (autologous) or from healthy donors (allogenic). Autologous stem cells are typically obtained from the patient’s own bone marrow or skin, cultured, and expanded in the lab before being reintroduced to the patient’s injured area. This approach avoids the potential for immune rejection since the cells come from the patient’s own body. However, the process of obtaining and culturing the cells can be time-consuming and may not be feasible for some patients. On the other hand, allogenic stem cells are obtained from healthy donors and can be prepared and stored for immediate use. This approach may offer a more convenient option for patients but comes with the risk of immune rejection, which can be mitigated through various techniques such as immunosuppressive therapy or genetic modification of the cells. The choice of using autologous or allogenic stem cells for skin injury treatment depends on various factors such as the patient’s condition, medical history, and the availability of appropriate stem cell sources. 

Exosomes are similar to stem cells in the sense that they can also be autologous or allogenic. However, exosomes have some advantages over stem cells in terms of their ease of isolation and potential for targeted delivery. Exosomes contain various bioactive molecules such as proteins, lipids, and nucleic acids. These molecules can mediate intercellular communication and influence various cellular processes such as proliferation, differentiation, and immune response. Allogenic exosomes can be obtained from healthy donors or exosome-producing cell lines. The isolation process can be relatively quick and straightforward, and the resulting exosomes can be modified to target specific cell types or tissues for improved therapeutic efficacy. Thus, both stem cells and exosomes hold great potential for skin injury treatment, and the choice between using autologous or allogenic sources depends on various factors such as availability, feasibility, and potential risks.

Several studies investigated the potential of stem cell therapy for the treatment of skin injury. For example, a study in rats found that adipose-derived stem cell transplantation significantly improved wound healing and reduced scarring in skin injuries [9]. Another study in mice found that BMSC transplantation improved wound healing and reduced inflammation in skin injuries [48].

Exosome therapy also showed promise for the treatment of SM-induced tissue injury. A study in rats found that exosomes derived from adipose-derived stem cells improved wound healing and reduced inflammation in skin injuries [49]. Another study in mice found that exosomes derived from BMSCs improved wound healing and reduced scarring in SM-induced skin injuries [50]. Although these studies suggest that stem cell and exosome therapy may be effective for the treatment of SM-induced skin injury, there is no clinical trial of these treatments for MS-induced disease. Further research is needed to determine the safety and efficacy of these treatments before use in humans.

### Mechanism of Stem Cell and Exosome Therapy

The mechanism of action for stem cell and exosome therapy in the treatment of SM-induced tissue injury is not fully understood. However, several mechanisms were proposed for how these therapies may promote tissue regeneration and reduce inflammation, using skin injury as an example (Figure 2).

Stem cells can differentiate into various cell types and release growth factors and cytokines that can promote tissue repair and regeneration [51]. When transplanted into injured tissue, stem cells can differentiate into the specific cell types needed for tissue repair and release factors that promote tissue healing [52,53,54,55].

Exosomes are small vesicles released by stem cells that contain various bioactive molecules, including proteins, nucleic acids, and lipids. Exosomes were shown to have anti-inflammatory and pro-regenerative effects in various tissues, including skin [6]. The bioactive molecules contained in exosomes can promote tissue repair, reduce inflammation, and modulate the immune response [56].

Several studies investigated the mechanisms underlying the effects of stem cell and exosome therapy in the treatment of tissue injury [19,47,57]. For example, a study in rats found that adipose-derived stem cells increased the expression of angiogenic factors, which promoted the formation of new blood vessels and reduced the expression of inflammatory cytokines in SM-induced lung tissue injuries [58]. Another study in rats found that adipose-derived stem cell exosomes promoted angiogenesis and reduced inflammation in skin injuries by activating the Wnt signaling pathway [59]. 

Currently, MSC or exosome therapy was used for SM-induced tissue injury in the eye and lung [19,47,57], or for other types of skin injury [20]. No study was reported on the stem cell and exosome therapy for SM-induced skin tissue injury. 

It is possible to use stem cell or exosome therapy for SM-induced skin injury. Although the mechanisms of stem cells or exosomes in the treatment of are not fully understood, these therapies showed promise in preclinical studies and may provide new avenues for the treatment of SM-induced tissue injury.

## 7. Therapeutic Outcomes of Stem Cell Therapy, and Exosome Therapy

Stem cell therapy and exosome therapy both showed potential for treating various diseases and injuries, including SM-induced tissue injury, and diabetic skin ulcers [60]. 

Therapeutic outcomes of stem cell therapy skin injury or wound healing are presented as an example:

In a preclinical study using a mouse model of skin injury, transplantation of adipose-derived stem cells (ASCs) resulted in faster wound healing, reduced inflammation, and increased angiogenesis compared to control mice [23].

A systematic review and meta-analysis of 10 clinical studies [61], involving a total of 453 patients with diabetic lower extremity vascular disease (LEVD), found that mesenchymal stem cell (MSC) therapy is more effective than conventional treatment in promoting wound healing and reducing scar formation. Patients who received MSC therapy had a higher rate of ulcer healing, reduced ulcers, and shorter complete healing time. Furthermore, MSC therapy was found to improve ankle-brachial index and transcutaneous oxygen pressure, indicating an improvement in blood supply. In addition, four of the included studies demonstrated that MSC therapy significantly increased the number of new collateral vessels. Importantly, no adverse events were reported in the MSC group, indicating a favorable safety profile. These findings suggest that MSC therapy holds great promise for promoting ulcer healing and improving blood supply in patients with diabetic LEVD, with no significant safety concerns.

Therapeutic outcomes of exosome therapy for skin wound:

A preclinical study investigated the effects of exosomes from human umbilical cord blood plasma (UCB-Exos) on wound healing and the underlying mechanism [50]. Results showed that UCB-Exos promoted wound healing, including accelerated re-epithelialization and enhanced angiogenesis. In vitro experiments also showed that UCB-Exos promoted the proliferation and migration of fibroblasts and enhanced the angiogenic activities of endothelial cells. MiR-21-3p was identified as a critical mediator in UCB-Exos-induced regulatory effects through inhibition of PTEN and SPRY1. These findings suggest that UCB-Exos could be a promising novel strategy for soft tissue wound healing. 

Evidently, both stem cell therapy and exosome therapy showed promising therapeutic outcomes in various preclinical and clinical studies, indicating their potential for treating SM-induced tissue injury and other skin diseases or injuries.

The benefits and limitations of stem cell therapy and exosome therapy, as listed in Table 2, are provided below.

Advantages of exosome therapy: Exosomes can be harvested from a variety of stem cell types and can be purified and standardized for therapeutic use [62]. Exosomes are small and can easily penetrate tissues, making them potentially more effective for targeted delivery of therapeutic molecules [63]. 

Exosomes can be pre-purified as an off-the-shelf product that can be used for SM-induced emergence injury in eyes, lungs, and skin. In addition, exosomes do not have the potential risks associated with stem cell transplantation, such as immune rejection or tumorigenesis [64].

Disadvantages of exosome therapy: The efficacy of exosome therapy may depend on the specific cargo of molecules contained in the exosomes, which can be influenced by factors such as the cell source and culture conditions [65]. The manufacturing process for exosomes can be challenging and may require specialized equipment and expertise [66].

Advantages of stem cell therapy: Stem cells have the potential to differentiate into various cell types and promote tissue regeneration through the release of growth factors and cytokines [67]. Stem cell therapy was shown to be effective in promoting tissue repair in preclinical and clinical studies [55]. Stem cell therapy can be used to treat a variety of tissue injuries, including SM-induced skin injury.

Disadvantages of stem cell therapy: Stem cell transplantation may carry the risk of immune rejection or tumor formation [68]. The differentiation potential of stem cells can lead to unwanted tissue formation or other complications if not properly controlled [69]. The manufacturing process for stem cell therapy can be complex and may require specialized facilities and expertise [69].

In summary, exosome therapy and stem cell therapy have their own unique advantages and disadvantages, and the choice between them will depend on the specific context and requirements of the treatment. More research is needed to fully understand the potential of each approach for the treatment of SM-induced tissue injury.

## 8. Therapeutic Outcomes, Benefits and Challenges of Combination Therapy

While stem cell therapy has the potential to regenerate damaged tissues, it takes time to yield enough autologous stem cells for tissue repair that is not suited for SM-induced emergence injury. Exosomes can be used as an off-the-shelf product that is immediately used for the emergence injury. Compared to stem cell therapy alone, exosome therapy has several advantages, including fewer safety concerns and better scalability. However, there are also potential pitfalls to exosome therapy, including the exosomes can be used for the emergence, heterogeneity of exosomes, limited understanding of mechanisms of action, and delivery challenges.

Combining stem cell therapy with exosome therapy may address some of these limitations, as it could be used to enhance tissue repair immediately with exosomes and replace damaged cells with stem cells a few weeks after expansion in culture. However, further research is needed to optimize this approach and evaluate its safety and efficacy in clinical trials. Noticeably, while there are significant challenges to treating SM-induced skin injury, stem cell and exosome therapy offers a promising new avenue for treatment that may ultimately improve patient outcomes.

Several studies on the therapeutic outcomes of this combination therapy for various tissue or organ injuries including SM-induced skin injury, brain, or heart tissue injury. In a preclinical study using rat model of SM-induced skin injury, co-administration of adipose-derived stem cells (ASCs) and exosomes derived from ASCs (ASC-Exos) resulted in significantly faster wound healing and increased angiogenesis compared to ASC or ASC-Exo alone [20]. In a preclinical study using a rat model of traumatic brain injury, co-administration of human umbilical cord mesenchymal stem cells (UCSCs) and exosomes derived from UCSCs (UCSC-Exos) resulted in improved cognitive function, reduced inflammation, and increased angiogenesis compared to UCSC or UCSC-Exo alone [70]. In a preclinical study using a mouse model of myocardial infarction, co-administration of cardiac progenitor cells (CPCs) and exosomes derived from CPCs (CPC-Exos) resulted in significantly greater cardiac function improvement and reduced fibrosis compared to CPC or CPC-Exo alone [71].

Obviously, combining stem cell therapy with exosome therapy showed promising therapeutic outcomes in various preclinical studies, indicating their potential for treating SM-induced skin injury and other tissue injuries. However, further clinical studies are needed to validate these findings and determine the optimal dosage and timing of combination therapy.

### Benefits of Stem Cell Therapy Combined with Exosomes

When combined, exosome therapy and stem cell therapy could potentially have a synergistic effect in the treatment of SM-induced injury in the eyes, lungs, and skin. The exosomes, as an off-the-shelf product, could help start the repair process immediately, while the stem cells, after expansion, could replace damaged cells over time. This combination therapy could potentially lead to better patient outcomes than either therapy alone (see Table 2), including: (1) enhanced therapeutic efficacy: preclinical studies showed that combining stem cells with their exosomes can improve therapeutic efficacy in various diseases and injuries, including myocardial infarction, traumatic brain injury, and SM-induced corneal and lung injury [70,71,72,73,74,75]. (2) Benefit in chronic tissue injury. (3) Reduced adverse effects: by using exosomes derived from stem cells, rather than the cells themselves, the risk of immunogenicity and tumorigenesis is reduced [65]. (4) Improved targeting and delivery: exosomes can target specific cells and tissues and can also cross biological barriers such as the blood–brain barrier [76]. This can enhance the delivery of therapeutic agents to the site of injury. (5) Longer-lasting impact: the combination results in an improved and more sustained therapeutic outcome, providing patients with a more practical and beneficial treatment option [71]. Thus, combining stem cell therapy with exosome therapy has the potential to improve therapeutic outcomes while reducing adverse effects and costs. However, further research is needed to determine the optimal combination therapy approach for different diseases and injuries.

Despite the potential benefits of stem cell and exosome therapy, there are still several challenges that need to be addressed before these therapies can be widely adopted in clinical practice. Some of these challenges include: (1) standardization and regulation: the lack of standardized protocols for the isolation, characterization, and storage of stem cells and exosomes remains a significant challenge [77]. Additionally, regulatory issues related to the use of stem cells and exosomes in clinical settings need to be addressed. (2) Limited understanding of mechanisms of action: while the therapeutic effects of stem cells and exosomes were demonstrated in preclinical studies, the mechanisms underlying these effects are not fully understood [78]. (3) Safety concerns: there is a risk of tumorigenicity and immunogenicity associated with the use of stem cells, and there is limited information on the long-term safety of exosome therapy [79]. (4) Scale-up and manufacturing: the production of large quantities of stem cells and exosomes for clinical use remains a significant challenge. Standardization and scalability of manufacturing processes are critical to the successful translation of these therapies into clinical practice [80]. (5) Cost: the cost of stem cell and exosome therapies can be high, which may limit their accessibility to patients [81].

Addressing these challenges will require continued research and development, as well as collaboration between researchers, clinicians, and regulatory agencies.

Optimizing these therapies for maximum effectiveness remains an ongoing challenge. One approach to optimizing stem cell therapy is to improve the survival, engraftment, and differentiation of stem cells after transplantation. Studies investigated various strategies to achieve this, including optimizing the timing and route of stem cell administration, using biomaterials or scaffolds to enhance cell survival and engraftment, and genetically modifying stem cells to improve their therapeutic potential [82,83,84]. Similarly, optimizing exosome therapy involves understanding and controlling key factors such as exosome isolation and characterization, dosing, and route of administration [25,85,86]. Additionally, improving the targeting and delivery of exosomes to the site of injury may enhance their therapeutic efficacy [69]. Various strategies, such as surface modification and functionalization, were explored to improve the targeting and delivery of exosomes [62,87,88]. Moreover, combining stem cell and exosome therapies may lead to a synergistic effect and provide a more effective treatment option for SM-induced tissue injury. However, further studies are needed to optimize the timing, dosing, and delivery of this combined therapy.

Although stem cell and exosome therapies hold promise for treating SM-induced skin injury, optimizing these therapies for maximum effectiveness remains an ongoing challenge. Further research is needed to improve the survival, engraftment, and differentiation of stem cells, and to better understand and control key factors in exosome therapy, in order to develop more effective treatments for this debilitating condition.

## 9. Directions for Future and Conclusions

In addition to the general future directions for stem cell and exosome therapy, there are some specific avenues for research in the context of SM-induced tissue injury. Some potential future directions for this area of research include: (i) identification of optimal cell sources: different types of stem cells may have varying regenerative potential and safety profiles in the context of SM-induced tissue injury. Further research is needed to identify the optimal cell sources for this condition, such as using urine-derived stem cells obtained from patients using non-invasive methods [51,89,90,91]. (ii) Development of personalized therapies: there may be individual differences in response to stem cell and exosome therapies. Personalized therapies, tailored to an individual’s specific needs and genetic makeup, could potentially improve outcomes. (iii) Evaluation of immune-modulatory effects: exosomes were shown to have immune-modulatory effects, which could be particularly relevant in the context of SM-induced tissue injury. Further research is needed to evaluate the potential of exosomes to modulate the immune response and improve outcomes in this condition. (iv) Comparison of different therapeutic approaches: stem cell and exosome therapy are just one of several emerging therapeutic approaches for SM-induced tissue injury. 

Future research should focus on optimizing stem cell, exosome therapy, tissue engineering, or immunotherapy, to identify the optimal treatment strategies for SM-induced tissue injury, and on conducting clinical trials to evaluate the safety and efficacy of these therapies. In addition, the development of new technologies, such as nanotechnology and gene editing, may also provide new avenues for the treatment of SM-induced skin injury. Ultimately, the development of effective therapies for SM-induced skin injury is crucial to improve the quality of life and outcomes of patients affected by this devastating condition.

In summary, SM-induced tissue injury is a devastating condition that can lead to both short-term and long-term complications. Stem cell or exosome therapy was used for SM-induced eye and lung injury; these therapies may be an alternative for SM-induced skin injury. Despite the development of various treatment options, including decontamination, wound care, and pharmacological agents, the challenges of managing this condition persist. One emerging therapeutic approach is stem cell and exosome therapy, which showed promising results in preclinical studies.

## Figures and Tables

**Figure 1 ijms-24-09947-f001:**
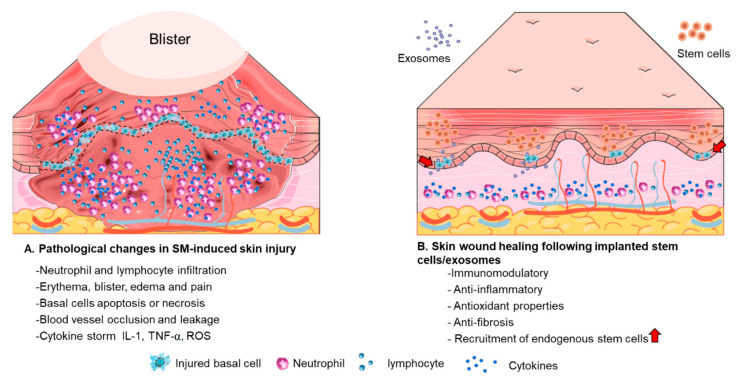
Stem cell or exosome therapy promotes skin tissue repair and regeneration.

**Figure 2 ijms-24-09947-f002:**
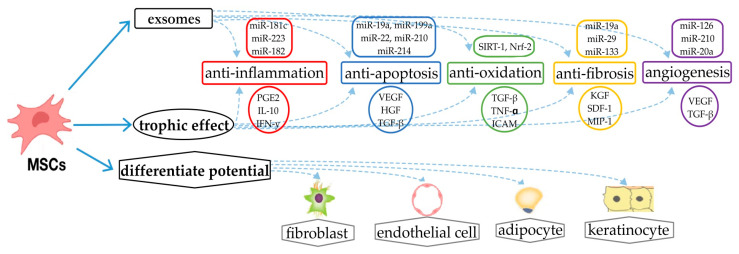
Stem cell and exosome therapy mechanism of action for treating skin injuries and diseases.

**Table 1 ijms-24-09947-t001:** Beneficial effect of adult stem cells on acute skin wound injury.

Cell Types	Outcomes	Refs.
BMSCs	Improved wound closure and extracellular matrix composition in mice	[10]
	Increased angiogenesis and re-epithelialization in mice	[11]
	Improved vascularity and decreased wound size in diabetic foot ulcers	[12]
AFSCs	Promoted wound closure in mice	[13]
	hAFSC exosomes accelerated wound healing and suppressed fibrotic scarring	[14]
UCSCs	Promoted wound closure, re-epithelialization, angiogenesis	[15]
CBSCs	Wound healing occurs as MSCs differentiate into keratinocytes in mice	[16]
	Both UCB MSCs and UCB-MSC-exos accelerated wound closure and reduced scarring in rats	[17]
PMSCs	Placental derived MSC accelerated wound healing in mice Placental MSCs and derived exosomes accelerate wound healing and increase regeneration of blood vessels	[18]
USCs	USCs promoted wound closure, re-epithelialization, angiogenesis, and collagen deposition	[19]

Abbreviations: Refs, references; BMSC, bone marrow stem cell; AFSC, amniotic fluid stem cell; UCSC, umbilical cord stem cell; CBSC, cord blood stem cell; PMSC, placental mesenchymal stem cell; USC, urine derived stem cell.

**Table 2 ijms-24-09947-t002:** Comprising Stem Cell Therapy and Exosome Therapy in SM-induced tissue injury.

	Stem Cells	Exosomes	Combination Therapy
Benefits	- Broadly studied - Cell differentiation ability - Self-renewal - Longer-lasting impact	- Mass-produced - Minimal invasive - Safety - No genetic manipulation - Easy storage and transportation	- Enhanced therapeutic efficacy - Benefit on chronic tissue injury - Reduced adverse effects - Improved targeting and delivery - Longer-lasting impact
Limitations	- Time-consuming, costly - Safety concerns for iPSCs - Ethical concerns for ESCs - Immune rejection for allogeneic	- Shorter half-life - Need multiple doses or continual release inventions - Limitation in severe tissue injury - Safety concern for iPSCs/ESCs-exosomes	- Standardization and regulation - Scale-up and manufacturing - Cost

Notes: ESCs—embryonic stem cells; iPSCs—induced pluripotent stem cells; combination: stem cell + exosome therapy.

## Data Availability

No new data was created in this review.

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
