# Peer review of "Therapeutic Benefits of Stem Cells and Exosomes for Sulfur-Mustard-Induced Tissue Damage"

_ijms, 2023, doi:10.3390/ijms24129947_

Round 1
Reviewer 1 Report
This is an insightful review, and looking forward to its publication; however, I have some suggestions that I hope the author will accept.
ADSC is a type of MSCs, not a parallel.
Note the consistency of writing between ADSC and ASC.
In addition to ADSCs, the role of other types of MSCs (bone marrow, amniotic fluid, umbilical cord, cord blood, placenta, epidermis, etc.) in wound healing should also be discussed (PMID: 17727462, 34344478, 36568306, 34235150, 33097078).
It still needs to check the writing.
没有。
Author Response
As recommended, we corrected the relationship of ADSCs and MSCs, from a parallel to a part and whole relationship.
We also added a new table (Table 1) to discuss the role of other types of MSCs (bone marrow, amniotic fluid, umbilical cord, cord blood, placenta, epidermis, etc.) The recommended references have been also included in the discussion.
Reviewer 2 Report
This manuscript reviews the current development of therapeutic benefit of stem cells and exosomes for skin damage by sulfur mustard. The review article is well-organized and written. Before considering this manuscript for publication, the authors should consider the following points in any revision as follows:
1. The whole manuscript only has 1 Figure. The authors should add more Figures for a better presentation of the paper by organizing several representative examples in the Figures when describing the corresponding text.
2. Standardize the format of the cited references before submitting to any journal. The page numbers were missed in the references. The authors should examine all the details of the manuscript carefully, which will undoubtedly reduce the workload of the editing manager of the Journal.
Author Response
1. As suggested, we added another figure (Figure 2) to present the mechanism of the treatment of MSCs and exosome therapy.
2. Format mistakes have been corrected. We have checked the format and details of the paper, especially the page numbers of the references as suggested.
Reviewer 3 Report
Dear Authors!
I found your review article regarding the therapeutic potential of stem cells and exosomes really interesting. It is well-organized, well-written and not so long maintaining the reading interest of readers.
As a minor points, I suggest to add two more figure into your manuscript clarifying 1) the stem cell and exosome therapy and 2) the future directions in this field.
Author Response
As suggested, we added another figure (Figure 2) to present the mechanism of the treatment of MSCs and exosome therapy. We also discussed the future direction in the last part of the paper.